

# Dynamical signatures of topological order in the driven-dissipative Kitaev chain

**Moos T. van Caspel[1], Sergio Enrique Tapias Arze[1] and Isaac Pérez Castillo[2,3⋆]**

**1** Institute of Physics and Delta Institute for Theoretical Physics,
University of Amsterdam, Amsterdam, The Netherlands
**2** Department of Quantum Physics and Photonics, Institute of Physics,
National Autonomous University of Mexico, Mexico
**3** London Mathematical Laboratory, 8 Margravine Gardens,
W6 8RH London, United Kingdom

⋆ isaacpc@fisica.unam.mx

## Abstract

We investigate the effects of dissipation and driving on topological order in superconducting nanowires. Rather than studying the non-equilibrium steady state, we propose a method to classify and detect dynamical signatures of topological order in open quantum systems. Bulk winding numbers for the Lindblad generator $\hat{\mathcal{L}}$ of the dissipative Kitaev chain are found to be linked to the presence of Majorana edge master modes – localized eigenmodes of $\hat{\mathcal{L}}$. Despite decaying in time, these modes provide dynamical fingerprints of the topological phases of the closed system, which are now separated by intermediate regions where winding numbers are ill-defined and the bulk-boundary correspondence breaks down. Combining these techniques with the Floquet formalism reveals higher winding numbers and different types of edge modes under periodic driving. Finally, we link the presence of edge modes to a steady state current.

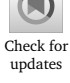

# 1   Introduction

Since its discovery, topological order is a subject that has attracted significant interest in condensed matter physics, with a number of research directions showing recent exciting activity [1, 2]. One of those is in the field of periodically driven systems, where research into Floquet topological insulators continues to reveal surprising properties [3–5]. Although these systems are inherently out-of-equilibrium, many defining features, such as long-range order and the bulk-boundary correspondence, remain present. Furthermore, entirely new physics emerges due to the periodic time evolution that results into additional topological invariants and phases.

Floquet theory has proven to be very effective in describing periodically driven quantum systems, which are used to engineer effective couplings leading to new states of matter [6]. However, for closed systems, particularly interacting ones, these periodic protocols often predict an unbounded heating, where they are driven to infinite temperature states at long times. More realistic models should therefore add some particle losses – inevitable in an experimental setting – which compete with the driving and lead the system to a Floquet steady state at finite temperature. Combining the Floquet formalism with the theory of open quantum systems results in an elegant framework for such driven-dissipative models [7, 8]. Unfortunately, this poses a major challenge for the study of topological order, typically defined for the system's ground state(s), as the non-unitary time evolution produces mixed quantum states. The topological classification of mixed states is an ongoing and contentious effort, with a number of open problems and conflicting findings [9–12].

Rather than focusing on topological properties of the steady state density matrix, the present work analyzes the full time evolution of an open quantum system. We draw inspiration from the study of topology in non-Hermitian Hamiltonians, which also lack a ground state. A flurry of recent work in this field shows that topological order can be inferred from their complex spectrum [13–19]. By writing the time evolution of a non-interacting open system in terms of non-Hermitian matrices, we apply some of these techniques and investigate which aspects of the topological order remain when a system is coupled to a bath in two cases:

firstly when the Hamiltonian part of the evolution is time independent, and secondly when it is subject to periodic driving.

As a case study, we consider the Kitaev chain, a spinless approximation of a superconducting nanowire with a topological phase that exhibits unpaired Majorana edge states [20]. The natural description in terms of Majorana fermions is well-suited for our treatment of the dissipation, which exploits the properties of Clifford algebras [21, 22].

The structure of this paper is as follows. In Section 2, we describe the model in detail and summarize its main equilibrium properties. In Section 3, we turn to our main goal of describing what happens to the Majorana edge modes and the bulk-boundary correspondence in the dissipative Kitaev chain. For the type of dissipation that we consider – a single, identical channel coupled to each site, with a parameter $\Delta$ interpolating between loss and gain – we find that the topological order of the Kitaev chain is preserved in a dynamical sense: edge modes now decay exponentially over time, but their existence is still guaranteed by symmetry. In addition, we find that the different topological phases are now separated by an intermediate region, where the band gap is closed and exceptional points appear [13]. This is the result of a spontaneously broken anti-unitary symmetry, similar to $\mathcal{PT}$ symmetry in non-Hermitian systems [23].

Finally, in Section 4, periodic driving is added to the system. In this case, the behavior of the system becomes even more complex as a consequence of Floquet resonances [24]. For example, one can now have a seemingly unlimited number of edge modes, which are split into two types [25, 26]. Moreover, in some regions of the parameter space, the edge modes can repel each other and break symmetries in a complex way. Instead of trying to describe and understand the full array of phenomena that can be seen in the driven-dissipative Kitaev chain, our goal is to demonstrate the promise of this approach and the richness of this kind of systems. We conclude in Section 5 with a number of open questions and possible future applications of our work.

## 2 The Kitaev chain with bulk dissipation

Originally studied in this context in [20], the Kitaev chain is a mean-field model of a 1D $p$-wave superconductor with spinless fermions, given by the Hamiltonian

$$H = -\sum_j \left( J c_j^\dagger c_{j+1} + \gamma c_j^\dagger c_{j+1}^\dagger + \text{H.c.} \right) - \mu \sum_j c_j^\dagger c_j, \tag{1}$$

where $J$ is the hopping amplitude, $\gamma$ the $p$-wave pairing and $\mu$ the chemical potential. Under periodic boundary conditions (PBC), the system has dispersion relation

$$\epsilon_k = \sqrt{4\gamma^2 \sin^2(k) + (2J\cos(k) + \mu)^2}. \tag{2}$$

The band gap closes at the critical point $\mu = \pm 2J$ and Kitaev showed that this corresponds to a topological phase transition. The easiest way of seeing this is by mapping to Majorana fermions:

$$w_{2j-1} = c_j + c_j^\dagger, \quad w_{2j} = i(c_j - c_j^\dagger), \tag{3}$$

resulting into the following Hamiltonian

$$H = -\frac{i}{2} \sum_j \left( (\gamma + J) w_{2j} w_{2j+1} + (\gamma - J) w_{2j-1} w_{2j+2} \right) - \frac{\mu}{2} \sum_j (1 + i w_{2j} w_{2j-1}). \tag{4}$$

The appearance of topological edge modes in the non-trivial phase, when considering open boundary conditions (OBC) with $N$ sites, becomes clear at the point $J = \gamma > 0$, $\mu = 0$. The Majorana fermions then form uncoupled dimers across neighboring sites, exactly like the limiting case of the prototypical Su-Schrieffer-Heger (SSH) model. At the edges, the modes $w_1$ and $w_{2N}$ disappear from the Hamiltonian entirely and therefore commute with $H$. Together they can be interpreted as a single delocalized fermionic mode with zero energy, which is protected from perturbations by symmetry as long as the gap does not close, resulting in a two-fold degeneracy of the ground state.

The symmetry of the system is crucial for topological order in 1D [27]. In the case of the Kitaev chain there is a combination of particle-hole and spatial inversion symmetry, which is often referred to as *chiral symmetry*. Spinless fermionic systems of this symmetry class are characterized by a $\mathbb{Z}_2$ topological invariant. Computing this so-called winding number from the system parameters can be done in a variety of ways. One possible way of doing so is the Zak phase, defined as the Berry phase over the full Brillouin zone:

$$\nu = \frac{i}{\pi} \int_{-\pi}^{\pi} \langle \psi_k | \frac{\partial}{\partial k} | \psi_k \rangle \, dk \mod 2 = \begin{cases} 0, & |\mu| > 2J \\ 1, & |\mu| < 2J \end{cases}, \tag{5}$$

where $|\psi_k\rangle$ is the eigenstate of $H$. A gauge transformation $|\psi_k\rangle \to e^{if(k)}|\psi_k\rangle$ for any (real, continuous) function $f(k)$ will contribute a multiple of $2\pi$ to the integral, and thus $\nu$ is only defined modulo 2. In the next section, we will discuss how the Zak phase is generalized to open systems.

## 2.1 Free Lindbladian time evolution

The time evolution of a quantum system in contact with a Markovian bath is described by the Lindblad master equation [28, 29]:

$$\hat{\mathcal{L}}\rho = \frac{d\rho}{dt} = -i[H, \rho] + \sum_m \left( L_m \rho L_m^\dagger - \frac{1}{2} L_m^\dagger L_m \rho - \frac{1}{2} \rho L_m^\dagger L_m \right), \tag{6}$$

where $\hat{\mathcal{L}}$ is the Liouvillian superoperator, $H$ is the system's Hamiltonian and the Lindblad operators $L_m$ encode the effect of the bath. If $H$ is quadratic and all $L_m$ are linear in fermionic operators, then the system is non-interacting and $\hat{\mathcal{L}}$ can be written in the following bilinear form:

$$\hat{\mathcal{L}} = \frac{1}{2} \sum_{i,j} \begin{pmatrix} \hat{c}_i^\dagger & \hat{c}_i \end{pmatrix} \mathbf{A}_{ij} \begin{pmatrix} \hat{c}_j \\ \hat{c}_j^\dagger \end{pmatrix} - A_0 \hat{\mathbb{1}}, \tag{7}$$

where $\hat{c}_j$ and $\hat{c}_j^\dagger$ are fermionic superoperators, satisfying $\{\hat{c}_i, \hat{c}_j^\dagger\} = \delta_{i,j}$ and acting on a Fock space of operators [21]. In particular, their action on the density operator is:

$$\hat{c}_j^\dagger \rho = \frac{1}{2} \left( w_j \rho + (\hat{P}\rho) w_j \right), \quad \hat{c}_j \rho = \frac{1}{2} \left( w_j \rho - (\hat{P}\rho) w_j \right), \tag{8}$$

where $\hat{P} = e^{i\pi \sum \hat{c}_j^\dagger \hat{c}_j}$ is the parity superoperator. One can show that $\hat{P}$ and $\hat{\mathcal{L}}$ commute, which implies that the Fock space is a direct sum of even and odd subspaces. The bilinear form (7) needs to be defined separately for each of these subspaces. For physical states, we can restrict ourselves to the even subspace, in which the $4N \times 4N$ *structure matrix* $\mathbf{A}$ takes the following block-triangular form [22]:

$$\mathbf{A} = \begin{pmatrix} -\mathbf{X}^\dagger & -i\mathbf{Y} \\ 0 & \mathbf{X} \end{pmatrix}, \tag{9}$$

$$\mathbf{X} \equiv -4i\mathbf{H} + 2\left(\mathbf{M} + \mathbf{M}^T\right), \quad \mathbf{Y} \equiv -4i\left(\mathbf{M} - \mathbf{M}^T\right), \tag{10}$$

where the *Hamiltonian matrix* **H** and the *bath matrix* **M** have dimensions $2N \times 2N$ and are defined by:

$$H = \sum_{i,j} H_{ij} w_i w_j,\tag{11}$$

$$L_m = \sum_j l_{m,j} w_j, \quad M_{ij} = \sum_m l_{m,i} l_{m,j}^*.\tag{12}$$

Here $w_j$ are Majorana fermions with anticommutator $\{w_i, w_j\} = 2\delta_{i,j}$, as defined by eq. (3). The overall shift $A_0 = \frac{1}{2}\,\text{Tr}\,\mathbf{X}$ in eq. (7) is required for trace conservation.

This formalism has a number of advantages. First, due to the block-triangular form of the structure matrix **A**, the spectrum of $\hat{\mathcal{L}}$ is completely determined by the real matrix **X**, specifically its $2N$ eigenvalues $\beta_i$. These are known as *rapidities* and have $\text{Re}(\beta_i) \geq 0$. If $\text{Re}(\beta_i) > 0$ for all $i$, then there is a unique non-equilibrium steady state $\rho_{ss}$ such that $\hat{\mathcal{L}}\rho_{ss} = 0$. Another advantage is that two-point functions in $\rho_{ss}$ are readily computed by solving the continuous Lyapunov equation [22] for the covariance matrix **C**:

$$\mathbf{X}^\dagger \mathbf{C} + \mathbf{C}\mathbf{X} = i\mathbf{Y}, \quad C_{ij} \equiv \text{Tr}(w_i w_j \rho_{ss}) - \delta_{i,j}.\tag{13}$$

Finally, by computing the (generalized) eigenvectors of **A**, we can construct *normal master modes* $\hat{b}_j$ and $\hat{b}_j'$, which act as free excitations on the operator Fock space. In this picture, the steady state is the vacuum for the master modes and, assuming **X** is diagonalizable, the Liouvillian takes the diagonal form

$$\hat{\mathcal{L}} = -\sum_j \beta_j \hat{b}_j' \hat{b}_j.\tag{14}$$

If **X** is a defective matrix, it is possible to write $\hat{\mathcal{L}}$ in a Jordan normal form with one or more nontrivial Jordan blocks [22]. The eigenmodes of the Liouvillian can then be constructed by acting with a combination of $\hat{b}_i'$ superoperators onto the steady state. Note that these modes are not density matrices, because they are traceless. To remain in the even sector of the operator Fock space, the number of master modes should be even.

To describe translationally invariant open systems we attach an identical, local bath to each site, represented by the Lindblad operator $L_j$. Then, the Hamiltonian and the Lindblad operators can be rewritten as follows:

$$H = \sum_{j=1}^{N} \sum_{\ell=-N/2}^{N/2} \begin{pmatrix} w_{2j-1} & w_{2j} \end{pmatrix} \mathbf{h}_\ell \begin{pmatrix} w_{2(j+\ell)-1} \\ w_{2(j+\ell)} \end{pmatrix}, \quad L_j = \sum_{\ell=-N/2}^{N/2} \underline{l}_\ell \cdot \begin{pmatrix} w_{2(j+\ell)-1} \\ w_{2(j+\ell)} \end{pmatrix},\tag{15}$$

where we have two Majorana modes per site $j$ and we consider periodic boundary conditions (PBC) $w_{j+2N} = w_j$. Now, one can perform a Fourier transform on $\ell$, to find the $2 \times 2$ matrices

$$\mathbf{h}(k) = \sum_{\ell=-N/2}^{N/2} e^{-ik\ell}\,\mathbf{h}_\ell, \quad \mathbf{m}(k) = \underline{l}(k) \otimes \underline{l}^*(k), \quad \underline{l}(k) = \sum_{\ell=-N/2}^{N/2} e^{-ik\ell}\,\underline{l}_\ell,\tag{16}$$

as functions of the quasi-momentum $k \in [-\pi, \pi]$. From these, the matrices $\mathbf{x}(k)$ and $\mathbf{y}(k)$ can be constructed according to eq. (10), which implies in turn that eq. (13) becomes a $2 \times 2$ matrix equation for the covariance matrix $\mathbf{c}(k)$ in Fourier space [30].

Applying eqs. (15) and (16) to the Kitaev chain Hamiltonian (4) yields:

$$\mathbf{h}(k) = \frac{1}{2} \begin{pmatrix} 0 & i(J\cos k + \frac{\mu}{2}) - \gamma \sin k \\ -i(J\cos k + \frac{\mu}{2}) - \gamma \sin k & 0 \end{pmatrix}.\tag{17}$$

For the bath, we choose general single-site Lindblad operators $L_j = \sqrt{g}(c_j + \Delta c_j^\dagger)$ where the real parameter $\Delta$ interpolates between pure gain ($\Delta \to \infty$) and loss ($\Delta \to 0$). The bath matrix then becomes:

$$\mathbf{m}(k) = \frac{g}{4}\begin{pmatrix} (1+\Delta)^2 & i(1-\Delta^2) \\ -i(1-\Delta^2) & (1-\Delta)^2 \end{pmatrix}. \tag{18}$$

The Lindbladian dynamics is therefore governed by the matrices:

$$\begin{aligned} \mathbf{x}(k) &= g(1+\Delta^2)\mathbb{1} + 2i\gamma \sin k\ \sigma_x - i(2J\cos k + \mu)\sigma_y + 2g\Delta\sigma_z, \\ \mathbf{y}(k) &= 2ig(1-\Delta^2)\sigma_y, \end{aligned} \tag{19}$$

where the $\sigma_\alpha$ are Pauli matrices. The corresponding spectrum of rapidities for $\mathbf{x}(k)$ is:

$$\beta(k) = g(1+\Delta^2) \pm \sqrt{4g^2\Delta^2 - 4\gamma^2\sin^2 k - (2J\cos k + \mu)^2}. \tag{20}$$

# 3 Topological order and edge modes

Previous studies on the topology of open quantum systems have largely focused on the steady state(s) [10,11,31–33], in analogy to topological order in the ground state of closed quantum systems. For mixed steady states, however, this is a complicated issue, since the conventional understanding of topological order does not necessarily hold for density matrices, resulting in a poor understanding of finite-temperature topological insulators [9].

For our study of the open Kitaev chain, the Lindblad operators $L_j$ break all symmetries of the Hamiltonian [34], resulting in a unique steady state independent of initial conditions. This implies that the ground state degeneracy of the closed system, caused by the zero-energy edge modes, does not survive the long-time limit [35]. However, we will see that the full dynamics of the system retains some of its original symmetries and, following studies of topology for non-Hermitian Hamiltonians [13,15] and driven systems [3], we can extract some topological properties by studying the spectrum and eigenmodes of the Liouvillian operator $\hat{\mathcal{L}}$.

## 3.1 Topological properties of $\mathbf{x}(k)$

Since the spectrum of $\hat{\mathcal{L}}$ is completely determined by the matrix $\mathbf{x}(k)$, it is worth exploring its symmetry properties. Actually, one can show that the matrices $\mathbf{A}$ and $\mathbf{X}$ share the same topological invariants (see Appendix C). For sake of clarity, let us start by considering the case of pure loss $\Delta = 0$. Here, $\mathbf{x}(k)$ and $\mathbf{h}(k)$ only differ by a constant shift and multiplicative factor, neither of which affects its eigenvectors. This implies that the winding number, given by eq. (5), of the closed system is well-defined and unchanged. One can similarly show that the same holds for the case of pure gain and for separate loss and gain channels on each site. The authors of [36] found a similar result for a dissipative SSH model, in which the topological band structure of the Liouvillian was identical to that of the Hamiltonian.

For $\Delta \neq 0$, even though the chiral symmetry of $\mathbf{x}(k)$ is broken, there remains an anti-unitary symmetry (AUS) [37,38], up to a constant shift:

$$\mathcal{S}(\mathbf{x}(k) - a_0\mathbb{1})\mathcal{S} = -(\mathbf{x}(k) - a_0\mathbb{1}), \tag{21}$$

where the operator $\mathcal{S} = K\sigma_x$ corresponds to a complex conjugation $K$ and a swapping between the even and odd Majorana modes, while the constant shift $a_0$ is related to the trace of $\mathbf{x}(k)$,

that is $a_0 = A_0/N = \frac{1}{2}\operatorname{Tr}\mathbf{x}(k) = 2g(1 + \Delta^2)$. Now, consider the biorthogonal left and right eigenvectors of $\mathbf{x}(k)$ [39]:

$$\mathbf{x}(k)\underline{u}_i(k) = \beta_i(k)\underline{u}_i(k), \quad \mathbf{x}^\dagger(k)\underline{v}_i(k) = \beta_i^*(k)\underline{v}_i(k),$$
$$\underline{v}_i^*(k) \cdot \underline{u}_j(k) = \delta_{ij}. \tag{22}$$

The symmetry (21) automatically implies that

$$\mathbf{x}(k)\mathcal{S}\underline{u}_i(k) = \big(2a_0 - \beta_i^*(k)\big)\mathcal{S}\underline{u}_i(k). \tag{23}$$

Now there are two options: either $\underline{u}_i(k)$ is also an eigenvector of $\mathcal{S}$, which implies that $\operatorname{Re}\beta_i(k) = a_0$ and $\operatorname{Im}\beta_i(k) \neq 0$; or $\underline{u}_i(k)$ and $\mathcal{S}\underline{u}_i(k)$ are the two different eigenvectors of $\mathbf{x}(k)$, with eigenvalues $\beta_i(k)$ and $2a_0 - \beta_i^*(k)$, respectively.

We say that the AUS is *unbroken* if the first case holds for all $k$ in the Brillouin zone. Otherwise, the symmetry is said to be spontaneously broken. Examples of the rapidity spectrum with unbroken and broken AUS can be seen in panels (a) and (b) of Figure 1. If $\operatorname{Re}\beta_i(k) = a_0$ for a subset of the Brillouin zone, as is the case in panel (b), then this subset is delimited by a pair of so-called *exceptional points* (EPs), marked with a red diamond in Figure 1. At these exceptional points the matrix $\mathbf{x}(k)$ becomes non-diagonalizable. Although the present analysis is reminiscent of $\mathcal{PT}$-symmetric Hamiltonians [23, 40], where the eigenvalues are real when the symmetry is unbroken, in our case the conjugation $K$ cannot be seen as time reversal as we are considering a non-unitary time evolution. Thus we refrain from labeling the AUS as a $\mathcal{PT}$ symmetry, although they are mathematically equivalent.

For the open Kitaev chain[1], the matrix $\mathbf{x}(k)$ has broken AUS whenever $2J - 2g\Delta \leq |\mu| \leq 2J + 2g\Delta$. This implies that between the two original phases of the Kitaev chain, an intermediate region of width $4g\Delta$ appears in which the band gap of $\mathbf{x}(k)$ is closed, as the two bands touch at the exceptional points. This is consistent with the findings of [13].

Without a chiral symmetry, it is not clear whether the $\mathbb{Z}_2$ topological invariant that characterizes the topology of the closed Kitaev chain is well defined, i.e. a quantized winding number. To investigate this, we generalize the winding number (5) to the biorthogonal basis (22), yielding:

$$\nu_i = \frac{i}{\pi}\int_{-\pi}^{\pi} \mathrm{d}k\, \underline{v}_i^*(k) \cdot \frac{\partial}{\partial k}\underline{u}_i(k), \tag{24}$$

where $\nu_i$ is the winding number associated to the band $i$ of $\mathbf{x}(k)$. In general, these numbers will not be real or quantized when chiral symmetry is broken. However, in the regime with unbroken AUS, one can show that their real part is quantized. To see this, note that the unbroken symmetry implies that:

$$\mathcal{S}\underline{u}_i(k) = \sigma_x\underline{u}_i^*(k) = e^{i\varphi_i(k)}\underline{u}_i(k), \quad \mathcal{S}\underline{v}_i(k) = \sigma_x\underline{v}_i^*(k) = e^{i\varphi_i(k)}\underline{v}_i(k), \tag{25}$$

where the eigenvalues must lie on the unit circle, since $\mathcal{S}$ is antiunitary. Using eq. (24), we can show that the winding number satisfies the following relation:

$$\nu_i = -\nu_i^* + \frac{1}{\pi}\int_{-\pi}^{\pi} \mathrm{d}k\, \frac{\partial}{\partial k}\varphi_i(k). \tag{26}$$

---

[1] In the present study, we assume $|\gamma| > \sqrt{\frac{g\Delta}{J}}J$ and $g\Delta < J$. The latter is warranted since we typically require weak dissipation for the Lindblad formalism to apply. If the former is not satisfied, then the gap can close at values of $k$ other than 0 or $\pi$, which increases the size of the AUS-broken regions. If $\gamma < g\Delta$, the topologically nontrivial phase will disappear completely. See Appendix A for more details.

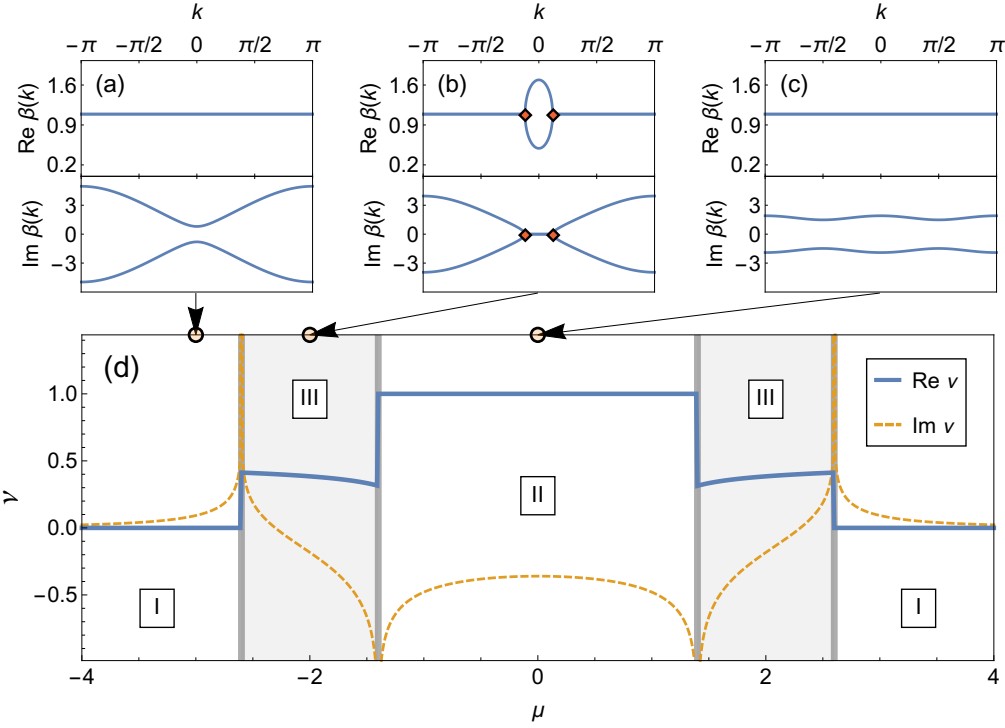

Figure 1: (a-c): The rapidity spectrum $\beta(k)$ for three values of $\mu$. Panel (a) at $\mu = -3$ and (c) at $\mu = 0$ both show unbroken AUS. Meanwhile, panel (b) is in the AUS-broken regime, with the two exceptional points marked by red diamonds. The bottom panel (d) shows the real (blue, solid) and imaginary (orange, dashed) parts of the winding number $\nu$ of $\mathbf{x}(k)$, with the topologically trivial phase (I), the nontrivial phase (II) and the intermediate regions (III) indicated. The other parameters are: $J = 1, \gamma = 0.8, g = 1.0, \Delta = 0.3$

If we demand the eigenvalue $e^{i\varphi_i(k)}$ to be single-valued within the Brillouin zone, then the last term must be an even integer. Therefore we must have $\nu_i = -\nu_i^* + 2n$, or equivalently $\text{Re}(\nu_i) = n$ for $n \in \mathbb{Z}$. Requiring gauge invariance again restricts this to $\mathbb{Z}_2$. From here on we will simply refer to the quantized real part as the winding number. Whether the imaginary part also has a physical interpretation in this context is not clear. We also note that $\sum_i \nu_i$ mod $2 = 0$, such that it is enough to study the winding number of only one of the two rapidity bands.

The resulting phase diagram of the open Kitaev chain can be found in Figure 1. As we can see in panel (d), we have now three phases: A trivial topological phase (denoted in the figure as $I$), a non-trivial topological phase (indicated as $II$) and, finally, an intermediate phase (indicated as $III$). In the non-trivial topological phase the AUS is unbroken and the real part of the winding number (for further details on its derivation see Appendix B) has quantized value 1, while in phase $I$ its value is zero. Let us now explore the effect of the bath on the bulk-boundary correspondence; whether a nonzero winding number still implies the existence of topologically protected edge modes, and what those modes might look like.

## 3.2 Bulk-boundary correspondence and its breakdown

While the bulk-boundary correspondence seems to generally fail for non-Hermitian systems [41], this is not the case for $\mathcal{PT}$-symmetric Hamiltonians [42]. As we have argued before,

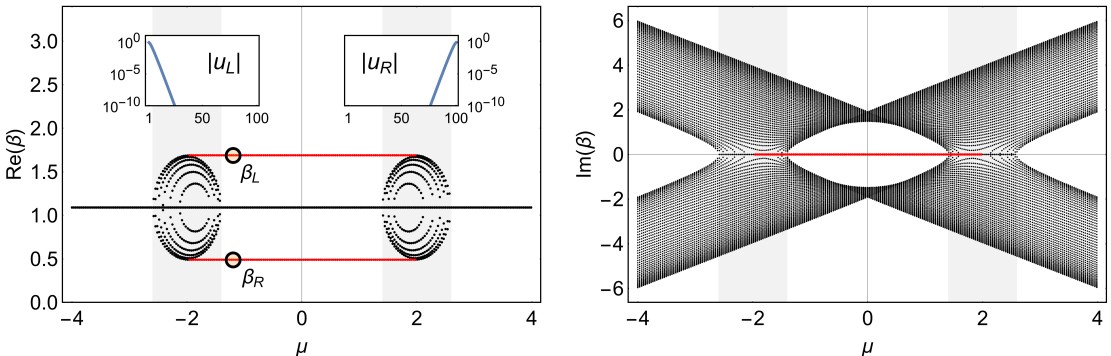

Figure 2: The real (left) and imaginary (right) parts of the spectrum of **X** with OBC, as a function of the chemical potential $\mu$. The localized edge modes with rapidity $\beta = g(1 \pm \Delta)^2$ are highlighted in red. In the shaded regions, the AUS is broken and the bulk band gap is closed. The insets show on a semi-log scale the localization of the eigenvectors $\underline{u}_L$ and $\underline{u}_R$, corresponding to the marked rapidities $\beta_L$ and $\beta_R$ at $\mu = -1.2$. The other parameters are: $N = 50, J = 1.0, \gamma = 0.8, g = 1.0, \Delta = 0.3$

the open Kitaev chain has an AUS that is mathematically equivalent to a $\mathcal{PT}$ symmetry, and therefore we expect bulk-boundary correspondence to hold in our case as well. That is to say, under OBC we expect two eigenvectors of **X** to be localized at the edges when $\nu = 1$, but none when $\nu = 0$. We have investigated this in two complementary ways. Firstly, we have analyzed the modes close to the gap in the continuum limit and studied the effect of domain walls which give rise to a localized edge mode (details of this approach can be found in Appendix D). Secondly, we have studied numerically the spectrum and eigenmodes of the $2N \times 2N$ matrix **X** with OBC. The resulting spectrum is shown in Figure 2.

Both of these analyses indicate the presence of two exponentially localized edge modes when $|\mu| < 2J$, i.e. in the nontrivial phase of the closed Kitaev chain. This can be clearly seen in Figure 2, where the red points mark the rapidities of the edge modes at $\beta = g(1 \pm \Delta)^2$, consistent with the derivation in Appendix D. The corresponding eigenvectors with localization length $\xi = 2\gamma/(2J - |\mu|)$ can be seen in the insets of that figure for one value of $\mu$. Comparing these results to the winding number of Figure 1(d), we conclude that the bulk-boundary correspondence holds as long the AUS is unbroken. In that case $\nu$ is quantized and accurately reflects the presence or absence of edge modes. In the intermediate region $2J - 2g\Delta < |\mu| < 2J + 2g\Delta$ where the AUS is broken, the bulk-boundary correspondence breaks down. Even though the band gap is closed, the topologically protected edge modes remain as long as $|\mu| < 2J$. Only when $\mu$ reaches $\pm 2J$, the real part of the bulk bands connects to the edge modes and they disappear, as shown in Figure 2. Interestingly, the winding number does not even show a discontinuity at this point.

## 3.3 Physical interpretation of edge master modes

An important question remains: what is the physical meaning of these edge modes? How could they be observed experimentally? In the closed Kitaev chain, Majorana edge modes can in principle be observed by scanning tunneling microscopy [43]. The localized eigenvectors of **X**, on the other hand, are not physical excitations on top of a ground state, and thus one cannot expect to detect them in the same way. Instead, they correspond to boundary master modes, as defined in eq. (14). In terms of the left and right eigenvectors of **X**, $\underline{v}_i$ and $\underline{u}_i$ respectively, and the covariance matrix **C**, these master modes $\hat{b}'$ and $\hat{b}$ are given by (see Appendix C for

details)

$$\hat{b}'_i = \underline{v}_i \cdot \underline{\hat{c}}^\dagger, \quad \hat{b}_i = \underline{u}_i \cdot \underline{\hat{c}} - \mathbf{C}\underline{u}_i \cdot \underline{\hat{c}}^\dagger. \tag{27}$$

A localized eigenvector then implies that the corresponding $\hat{b}'$ is localized too, while localization of $\hat{b}$ also requires that there are no long-range correlations in the steady state.

Now, define $\hat{b}'_L$ and $\hat{b}'_R$ to be the two edge master modes, corresponding to the localized eigenvectors. In the limiting case $\mu = 0$ and $J = \gamma$, where the Majorana edge modes decouple completely from the bulk, we can analyze the system's behavior exactly. The edge master modes then take the simple form: $\hat{b}'_L = \hat{c}^\dagger_1, \quad \hat{b}'_R = \hat{c}^\dagger_{2N}$. All the other master modes are restricted to the bulk, since they must be orthogonal to the edge modes. To understand the physical consequences of this type of localization, let us consider the expectation value of the edge occupation number in this limit:

$$\langle \mathcal{O}(t) \rangle = \mathrm{Tr}(d^\dagger_0 d_0 \rho(t)), \quad d^\dagger_0 \equiv \frac{1}{2}(w_1 + i w_{2N}). \tag{28}$$

Taking as initial condition the (pure) ground state with the edge mode occupied, i.e. $\langle \mathcal{O}(0) \rangle = 1$, we can then turn on the dissipation and study the time evolution. Since the edges are decoupled, only the decay mode $\hat{b}'_L \hat{b}'_R \rho_{ss}$ contributes to this expectation value, with a decay rate $2g(1 + \Delta^2)$ given by its eigenvalue. In the steady state, we find $\lim_{t \to \infty} \langle \mathcal{O}(t) \rangle = 0$, as can be computed using eq. (13). Therefore the time evolution is simply given by:

$$\langle \mathcal{O}(t) \rangle = e^{-2g(1+\Delta^2)t}. \tag{29}$$

In other words, the Majorana zero modes of the Kitaev chain are still present in the system, but acquire an exponential decay. These results are in agreement with those found in [35] in a similar context.

If we look at the evolution of *local* observables, such as the single-site fermionic occupation number, then the fact that the edge master modes decay at different rates becomes apparent. This is shown in Figure 3, where we plot the time evolution of the local density at the edges (dashed red line for the left edge, and solid blue line for the right one) and at the center of the chain (green dash-dot line) (details of this derivation are described in Appendix E). The occupation at the right edge of the system approaches its steady state value more slowly than the occupation number at the left edge. The difference between these decay rates is controlled by the parameter $\Delta$, responsible for the split between the real parts of the edge mode rapidities, and it vanishes for $\Delta = 0$. In the middle of the chain, neither of the edge modes contributes to the single-site occupation and the decay rate is halfway between that of the two edges. Since the decay modes in the bulk all have complex rapidities, there is also an oscillating part to the expectation values.

Note that, since the edge modes have rapidities $\beta = g(1 \pm \Delta)^2$, something interesting happens at $\Delta = \pm 1$: one of the rapidities vanishes. This means that one of the pair of edge modes does not decay and the steady state becomes degenerate. Intuitively, this makes sense: the Lindblad operators $L_j$ are now proportional to one of the Majorana operators $w_{2j}$ or $w_{2j-1}$, and will consequently leave one of the Majorana edge modes isolated from the bulk. However, this degenerate steady state $\rho'_{ss}$ is restricted to the odd sector of the operator Fock space and, as such, it can never be reached from a physical state in the even sector, implying that is not a true degeneracy. One way around this might be to consider a trivial domain ($|\mu| > 2J$) within the bulk of a nontrivial Kitaev chain with OBC. With two pairs of edge modes at the four different domain walls, it is possible to construct a steady state degeneracy in the even sector, where two Majorana edge modes decay and two remain.

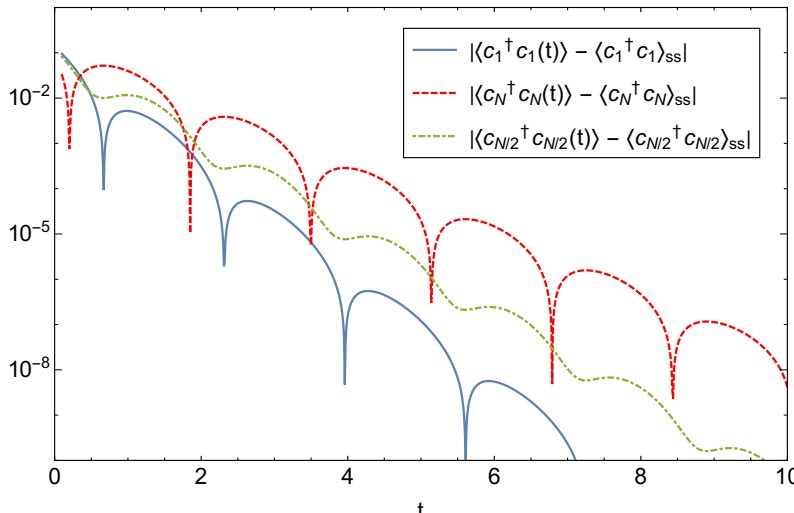

Figure 3: Approach to the steady state value of local occupation numbers at the left edge (blue, solid), right edge (red, dashed) and center (green, dash-dot) of the dissipative Kitaev chain with OBC. The absolute value is shown on a semi-log scale, in order to highlight the exponentially decaying envelopes and their different decay rates, caused by the presence of topological edge modes. This is in the limiting case of decoupled Majorana dimers, with parameters: $N = 100, J = 1.0, \gamma = 1.0, \mu = 0, g = 1.0, \Delta = 0.3$

## 3.4 Steady state expectation values

As we have already argued (see Figure 3), the presence of edge modes affects the rates at which some observables approach their steady state expectation values. A logical next step would be to determine whether the steady state expectation values themselves also reflect some topological order. Although the Majorana edge modes decay under influence of the bath, they might leave some signature in the steady state covariance matrix. By numerically solving the Lyapunov equation (13) with OBC, we can extract the expectation values of various observables. In particular, it is possible to study the expectation value of the occupation number at the edges, as a function of $\mu$, to see whether the phase transition could be detected through these observables. In order to compare the occupation at the edges relative to the bulk, we define the edge occupation ratios:

$$r_L = \frac{\langle c_1^\dagger c_1 \rangle - \langle c_{N/2}^\dagger c_{N/2} \rangle}{\langle c_{N/2}^\dagger c_{N/2} \rangle}, \qquad r_R = \frac{\langle c_N^\dagger c_N \rangle - \langle c_{N/2}^\dagger c_{N/2} \rangle}{\langle c_{N/2}^\dagger c_{N/2} \rangle} . \tag{30}$$

We expect this quantity to be nonzero even in the absence of topological edge modes, due to the boundary conditions. However, when plotted as a function of the chemical potential, these ratios have a clear peak in the topologically nontrivial phase. This can be seen in the panels (a) and (b) of Figure 4, where we show the numerical results for two different values of the dissipation strength $g$. In the limit $g \to 0$, there are sharp kinks at the critical points $\mu = \pm 2J$ and the left and right edge show identical behavior. As the dissipation strength is increased, the phase transition is smoothened out and, for $\Delta \neq 0$, there is an increased asymmetry between the left and right edge, as can be verified in panel (b).

This asymmetry can be seen as an unequal charge build-up on the edges of the system. Therefore one might expect that, if we consider PBC instead of OBC, there would be a current

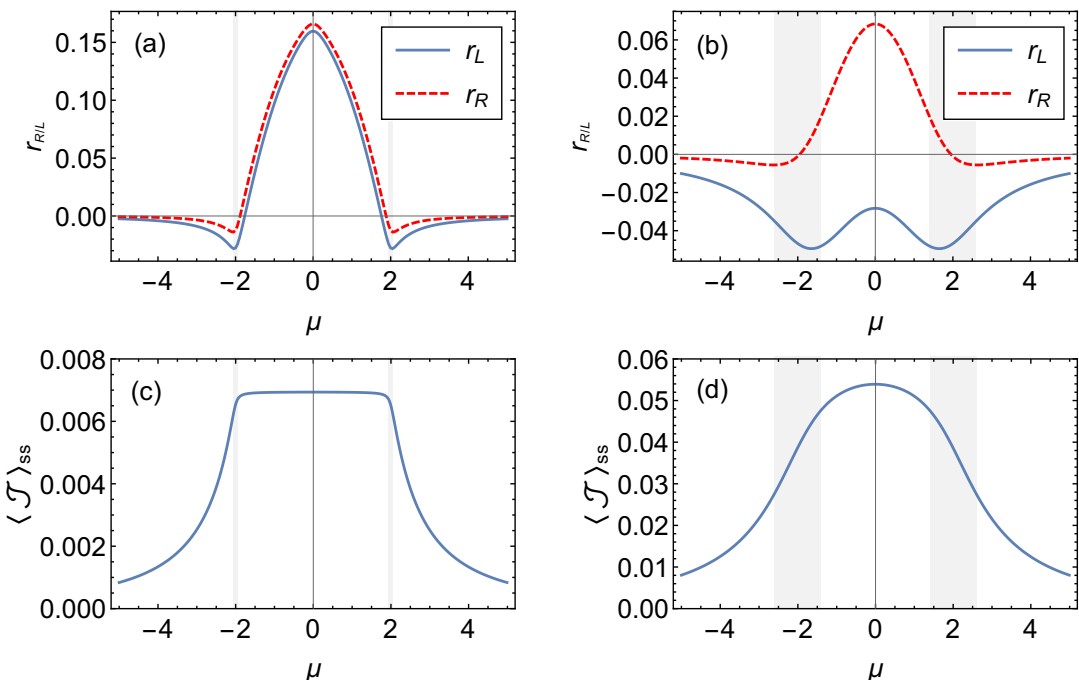

Figure 4: Top: edge occupation ratios in the steady state, as defined in eq. (30), under OBC with $N = 100$. (a) shows weak dissipation ($g = 0.1$) while (b) has stronger dissipation ($g = 1.0$). The latter shows a smoother transition and a large asymmetry between the left and right edge. Bottom: steady state current $\langle \mathcal{J} \rangle$ under PBC, with weak (c) and strong (d) dissipation. Shaded regions have broken AUS. The remaining parameters are: $J = 1$, $\gamma = 0.8$, and $\Delta = 0.3$.

flowing in the steady state. One way to define the electronic current is:

$$\langle \mathcal{J} \rangle = \frac{i}{N} \sum_j \left( \langle c_j^\dagger c_{j+1} \rangle - \langle c_{j+1}^\dagger c_j \rangle \right) = -\frac{1}{N} \sum_j \mathrm{Im} \langle c_j^\dagger c_{j+1} \rangle . \tag{31}$$

The steady state expectation value of this operator can be seen in panels (c) and (d) of Figure 4, again for the same two values of $g$. As with the edge occupation ratio, there are sharp kinks at the transitions between the topologically trivial and the nontrivial regimes when $g \to 0$, and stronger dissipation smoothens out this transition. The expectation value of the current operator also increases with the strength of the dissipation. The current reverses direction if we change the sign of $\Delta$ and vanishes at $\Delta = 0$.

We stress that these observables are not order parameters: in the open system, there is no sharp transition or discontinuity at the critical value of $\mu$. However, in the limit $g \to 0$, kinks appear and the derivatives become discontinuous. Furthermore, we have verified the presence of a steady state current under PBC, which appears to susceptible to topological order. This may seem counter-intuitive at first glance, but note that the $p$-wave pairing terms in the Hamiltonian (1) are not symmetric under spatial inversion, as they pick up a minus sign. It is therefore natural that the steady state does not have this symmetry.

## 4 Higher winding numbers in the driven-dissipative Kitaev chain

Driven closed quantum systems with chiral symmetry have been shown to accommodate multiple pairs of edge modes, of which one can distinguish two different types [3, 25]. Topological

phases are in this case characterized by a $\mathbb{Z} \times \mathbb{Z}$ invariant. A relevant question is then what happens to these properties when a bath is coupled to a driven system. This has been somewhat explored in the $XY$ Heisenberg spin chain [8], which maps to the Kitaev chain by a Jordan-Wigner transformation [44]. These studies were further extended to identify the existence of Floquet Majorana edge modes [45].

We will now explore what remains of the structure previously unveiled in the open Kitaev chain, once driving is added. In particular we will look into greater detail at the non-equilibrium properties of these Floquet Majorana edge modes and their classification into two types as described in [25, 26]. We show that the combined effects of the driving and the bath induce regions of higher winding number which are separated by intermediate phases of broken AUS.

## 4.1 Topology and the Floquet formalism

As it is well known, a closed periodically driven system, with $H(t + T) = H(t)$, can be studied using Floquet theory [46]. In this case, the stroboscopic time evolution is generated by the Floquet Hamiltonian $H_F(t_0)$, defined as:

$$\mathcal{U}(t_0 + T, t_0) = \mathcal{T} e^{-i \int_{t_0}^{t_0+T} \mathrm{d}\tau H(\tau)} = e^{-iH_F(t_0)T} .$$

(32)

Similarly to what happens in a periodic lattice, where quasi-momenta can be restricted to the Brillouin zone, the eigenvalues $\epsilon_F$ of $H_F(t_0)$ are only defined up to multiples of $2\pi/T$, resulting in a Floquet Brillouin zone with $\epsilon_F \in [-\pi/T, \pi/T)$. For this reason these eigenvalues are referred to as quasi-energies. Note that $H_F(t_0)$ depends on the initial time $t_0 \in [0, T)$ for which the Floquet propagator (32) is defined, but the quasi-energy spectrum does not [46].

The periodicity of quasi-energies in a driven two-band system with chiral symmetry results in the presence of *two* band gaps, one at $\epsilon_F = 0$ and another one at $\epsilon_F = \pi/T$. In each band, topologically protected edge modes may arise, characterised by their own winding number. It has been shown that the driven Kitaev chain has a $\mathbb{Z} \times \mathbb{Z}$ topological invariant [3, 25] – an expected result, since the periodic driving introduces an additional $\mathcal{S}_1$ manifold.

In order to see this, we need to determine whether the Floquet Hamiltonian $H_F(t_0)$ inherits the chiral symmetry of the instantaneous Hamiltonian $H(t)$. For sake of clarity we will consider periodic quenches of the chemical potential $\mu$, alternating between the values $\mu_1$ during a time $t_1$, and $\mu_2$ during a time $t_2$, with $T = t_1 + t_2$ being the driving period. It is easy to check that the Floquet Hamiltonian in this scenario has chiral symmetry only at two particular points $t'$ and $t''$ within each period, precisely where the driving protocol is time-reversal invariant - in our case, at the center of the two time plateaus. Let us denote the corresponding Floquet Hamiltonians as $H_{F'} = H_F(t')$ and $H_{F''} = H_F(t'')$, with

$$e^{-iH_{F'}T} = e^{-iH_1 t_1/2} e^{-iH_2 t_2} e^{-iH_1 t_1/2} ,$$
$$e^{-iH_{F''}T} = e^{-iH_2 t_2/2} e^{-iH_1 t_1} e^{-iH_2 t_2/2} .$$

(33)

These two effective Hamiltonians share the same spectrum, but can have different bulk winding numbers $\nu'$ and $\nu''$, since they are related by the $k$-dependent unitary transformation $G = e^{-iH_2 t_2/2} e^{-iH_1 t_1/2}$. Using this transformation, one is able to derive that:

$$\frac{\nu' - \nu''}{2} = \frac{i}{2\pi} \int_{-\pi}^{\pi} \mathrm{d}k \, \langle \psi_{F'} | G^{\dagger}(\partial_k G) | \psi_{F'} \rangle \equiv \nu_\pi ,$$

(34)

where $|\psi_{F'}(k)\rangle$ are eigenstates of $H_{F'}$. One can show that this $\mathbb{Z}$ topological invariant counts the number of Floquet Majorana edge modes with quasi-energy $\epsilon_F = \pi/T$ under open boundary conditions [25, 47]. Note that, unlike the winding number of undriven systems given by

eq. (5), $\nu_\pi$ does not have to be defined modulo 2 in order to be gauge invariant, which implies that more than two Floquet Majorana edges modes can accumulate at the same band gap. Similarly, the other $\mathbb{Z}$ invariant, which is given by the combination $\nu_0 \equiv (\nu' + \nu'')/2$, counts the edge modes with $\epsilon_F = 0$. Therefore, by driving the system, one can get multiple orthogonal edge modes at each end of the chain in contrast to a single one in the undriven case.

## 4.2 Lindblad-Floquet theory

To see how the two new topological invariants are affected by the presence of a bath, we first need to generalize Floquet theory to encompass the Lindbladian time evolution. For a time-dependent Liouvillian, the time-evolution superoperator over one period is $\hat{\mathcal{U}}(t_0 + T, t_0) = \hat{T} \exp\left(\int_{t_0}^{t_0+T} d\tau \hat{\mathcal{L}}(\tau)\right)$. Whether this can be written in terms of a Floquet Liouvillian $\hat{\mathcal{L}}_F$ depends on the details of the system [48], but in our case it does not pose a problem and therefore we write

$$\hat{\mathcal{U}}(t_0 + T, t_0) \equiv e^{\hat{\mathcal{L}}_F(t_0)T}, \tag{35}$$

where $\hat{\mathcal{L}}_F(t_0)$ can be expressed, as done in eq. (7), in terms of a effective structure matrix $A_F$. Next, to preserve the antiunitary symmetry (21) of the dissipative Kitaev chain we follow the procedure described in eq. (33). We pick as the starting time $t_0$ one of the two time-reversal invariant points $t'$ and $t''$ in the two-step driving protocol and write:

$$e^{A_{F'}T} = e^{A_1 t_1/2} e^{A_2 t_2} e^{A_1 t_1/2} = \begin{pmatrix} e^{-X_{F'}^\dagger T} & -i\mathbf{Q}' \\ 0 & e^{X_{F'}T} \end{pmatrix}, \tag{36}$$

$$e^{X_{F'}T} = e^{X_1 t_1/2} e^{X_2 t_2} e^{X_1 t_1/2}. \tag{37}$$

The computation of the off-diagonal block $\mathbf{Q}'$ is somewhat complicated and shown in Appendix F. Similar equations define the matrices $A_{F''}$, $X_{F''}$, and $\mathbf{Q}''$ for the Floquet Liouvillian $\hat{\mathcal{L}}_{F''}$ at time $t''$. Our main focus will be on the properties of $X_{F'}$ and $X_{F''}$ as their eigenvectors determine the topological properties of the driven-dissipative system. Their eigenvalues, which we refer to as *quasi-rapidities*, are defined up to a multiple of $2\pi i/T$ and can therefore be restricted to the first Floquet Brillouin zone: $-\pi/T < \text{Im}(\beta) \le \pi/T$. For PBC, translational invariance allows us to decompose the matrices $X_F$ in Fourier components of $2 \times 2$ matrices $x_F(x)$. It turns out that if the Fourier components $x_1(k)$ and $x_2(k)$ satisfy the AUS condition (21), so do the matrices $x_{F'}(k)$ and $x_{F''}(k)$, due to the symmetric composition of the exponentials in eq. (37). Therefore, from the result we discussed for the undriven case, we can expect regions in parameter space where the AUS is unbroken and the winding numbers $\nu_0$ and $\nu_\pi$ are quantized, separated by intermediate regions containing exceptional points.

As we will see, in this case, the phase diagram turns out to be much richer: aside from the known phases of the undriven Kitaev chain described in Sec. 2, there are new transitions induced by Floquet resonances. Such resonant regimes are characterized by long-range order [8], nonlocal Floquet Hamiltonians [24] and the possibility of high winding numbers [45].

To obtain an expression for the spectrum of $x_{F'}(k)$ and $x_{F''}(k)$, as well as the topological winding numbers $\nu_0$ and $\nu_\pi$, we proceed as follows. Using Euler's formula we write

$$e^{\frac{1}{2}t_1 x_1(k)} \equiv e^{\frac{1}{2}t_1 a_0}\left(m_0 \mathbb{1} + i\underline{m} \cdot \underline{\sigma}\right), \quad e^{t_2 x_2(k)} \equiv e^{t_2 a_0}\left(n_0 \mathbb{1} + i\underline{n} \cdot \underline{\sigma}\right), \tag{38}$$

$$e^{T x_{F'}(k)} \equiv e^{T a_0}\left(p_0 \mathbb{1} + i\underline{p} \cdot \underline{\sigma}\right) = e^{T a_0}(m_0 \mathbb{1} + i\underline{m} \cdot \underline{\sigma})(n_0 \mathbb{1} + i\underline{n} \cdot \underline{\sigma})(m_0 \mathbb{1} + i\underline{m} \cdot \underline{\sigma}), \tag{39}$$

with $\underline{\sigma} = (\sigma^x, \sigma^y, \sigma^z)$, and where $\underline{m}$, $\underline{n}$ and $\underline{p}$ are complex valued vectors. Once again we have $a_0 = g(1 + \Delta^2)$. After some algebraic manipulation, one can show that

$$
\begin{aligned}
p_0 &= m_0^2 n_0 - n_0\, \underline{m} \cdot \underline{m} - 2m_0\, \underline{m} \cdot \underline{n}\,, \\
\underline{p} &= (2m_0 n_0 - 2\underline{m} \cdot \underline{n})\underline{m} + (m_0^2 + \underline{m} \cdot \underline{m})\underline{n}\,.
\end{aligned}
\tag{40}
$$

The quasi-rapidity spectrum of $\mathbf{x}_{F'}(k)$ is then given by $\beta_F(k) = a_0 \pm i\cos^{-1}(p_0)/T$, and its winding number is the same as that of $\underline{p}(k) \cdot \underline{\sigma}$. Moreover, the AUS guarantees that $p_z$ is real while $p_x$ and $p_y$ are purely imaginary. Starting from eq. (24) and following the derivation in Appendix B, we finally obtain an expression for the winding number:

$$
\nu' = \frac{1}{2\pi} \int_{-\pi}^{\pi} \left( 1 \pm \frac{p_z}{\sqrt{p_x^2 + p_y^2 + p_z^2}} \right) \frac{p_x \frac{\partial}{\partial k} p_y - p_y \frac{\partial}{\partial k} p_x}{p_x^2 + p_y^2}\, \mathrm{d}k\,.
\tag{41}
$$

The same calculation can be done for $\mathbf{x}_{F''}(k)$ at the other time-reversal invariant point to obtain the winding number $\nu''$. By adding and subtracting the resulting winding numbers, we finally find the two topological invariants $\nu_0$ and $\nu_\pi$ of the driven-dissipative system.

## 4.3 Numerical results

In order to build some intuition for the problem, let us first consider the infinite-frequency limit ($T \to 0$), for which the stroboscopic time evolution is governed by the average Liouvillian [7]. In this case, that is simply the dissipative Kitaev chain with $\mu = (\mu_1 t_1 + \mu_2 t_2)/T$. As the driving period is increased, the Floquet Brillouin zone shrinks. Eventually, the quasi-rapidity bands reach the edges of the Floquet Brillouin zone, at which point the gap at $\pi/T$ closes. As we further increase $T$, we first encounter a finite intermediate region with broken AUS, after which the gap reopens and the first Floquet resonance occurs. This phase is characterized by avoided crossings of the quasi-rapidity bands and nonlocal hopping in the effective generator of (stroboscopic) time evolution [24]. According to the bulk-boundary correspondence, if we were to use OBC, an edge mode would be present inside the reopened gap. For large enough $T$, the resonances become more frequent and start to overlap, i.e. several resonances can be present at once. This is the mechanism that leads to multiple edge modes per gap.

The behavior described above can be seen in the bottom panel of Figure 5, which shows the imaginary part of the quasi-rapidities as a function of the period. As the Floquet resonances accumulate, the number of edge modes in each gap can be clearly seen from the real part of the spectrum, shown in the middle panel of the same figure. Away from the AUS-broken regions, these numbers neatly coincide with the two bulk winding numbers $\nu_0$ and $\nu_\pi$, computed numerically from eq. (41) and shown in the top panel of Figure 5. The real part of the quasi-rapidities, which arises purely as a consequence of non-Hermiticity, is more than a visual aid to distinguish the different edge modes: in the absence of chiral symmetry, it provides the mechanism that prevents two edge modes from recombining and scattering, even when they occupy the same edge and the same gap. As long as the real parts of the quasi-rapidities are different, the corresponding modes cannot hybridize.

However, the hybridization of two edge modes can occur when they have the exact same quasi-rapidity. At that point, the imaginary parts of the pairs of edge modes can repel each other, resulting in two pairs with the same real part. These hybridized edge modes have an imaginary part that is neither 0 nor $\pi/T$ and are not counted by the winding numbers, although they are still localized at the edges. Whether they are also topologically protected in some way is an interesting open question. If they are, then this can be seen as further violation of the bulk-boundary correspondence. These hybridized edge modes can be observed for certain values of $T$ as additional black lines in the real part of the spectrum in the middle panel of

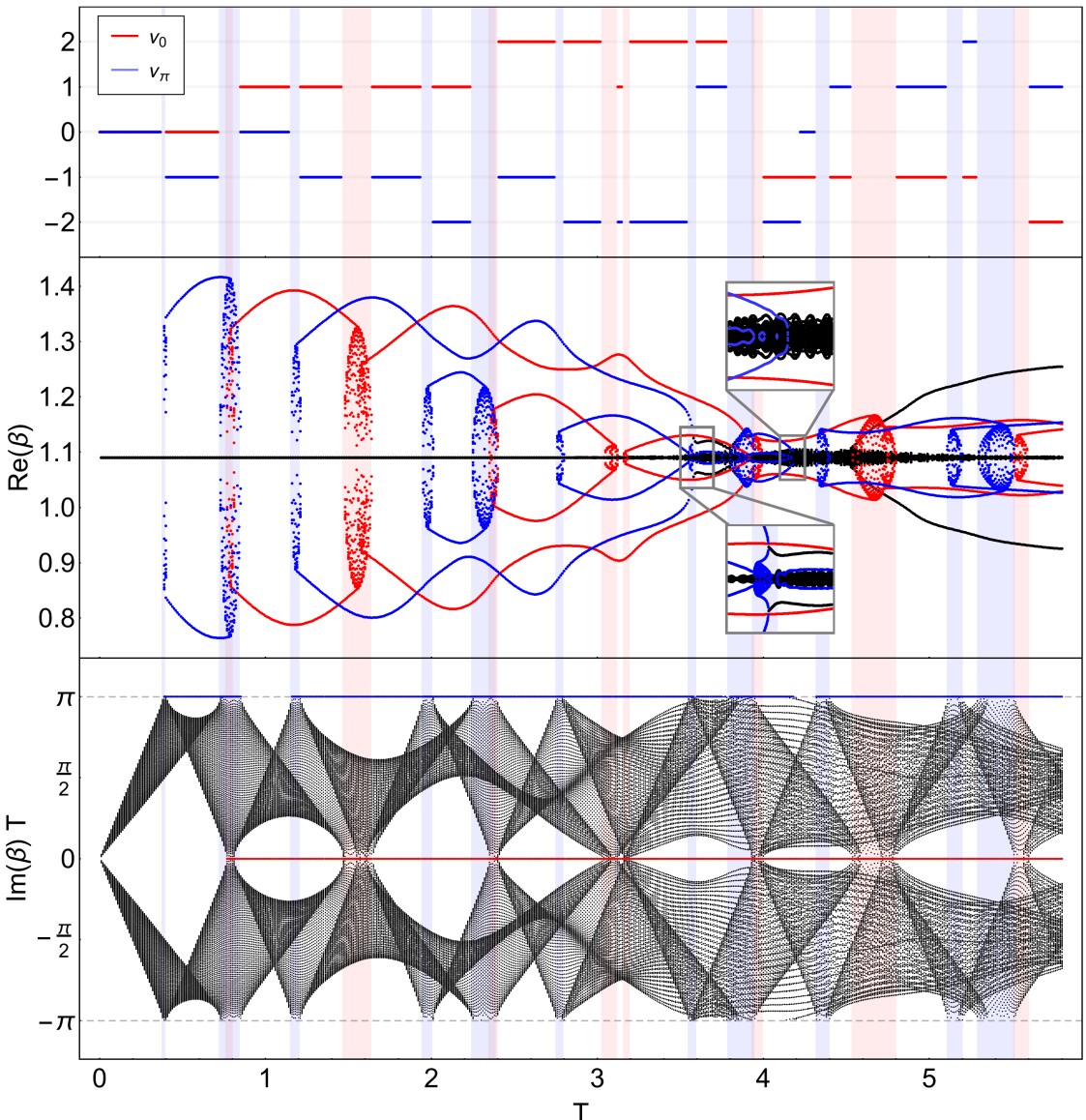

Figure 5: Winding numbers (top panel) and quasi-rapidities for OBC (middle and bottom panels) as a function of the driving period $T$. In all three panels, shaded regions correspond to broken AUS, with the bulk band gap closing either at $\beta = a_0$ (red background) or $\beta = a_0 + i\pi/T$ (blue background). The winding numbers $\nu_0$ and $\nu_\pi$ are shown only outside these regions. The real and imaginary parts of the quasi-rapidity spectrum under OBC are shown in the bottom two panels. Red dots correspond to quasi-rapidities with $\text{Im}(\beta) = 0$, and blue dots to $\text{Im}(\beta) = \pi/T$. The edge modes corresponding to each of these gaps can be seen clearly in the real part of the spectrum as lines connecting different AUS-breaking 'bubbles'. The number of pairs of red and blue edge modes is given by $|\nu_0|$ and $|\nu_\pi|$ respectively. The insets in the middle panel show magnifications of regions where edge modes hybridize. The parameters are: $N = 100$, $J = 1$, $\gamma = 0.8$, $g = 1.0$, $\Delta = 0.3$, $\mu_1 = 12$, $\mu_2 = 0$, and $t_1 = t_2$.

Figure 5. They are present around $T \approx 3.7$ (enlarged in an inset), as well as for $T > 4.5$. A similar phenomenon seems to happen around $T \approx 4.2$, where $\nu_\pi$ jumps from $-2$ to $0$ while the bulk band gaps do not close. In the real part (also enlarged in an inset), we see two pairs

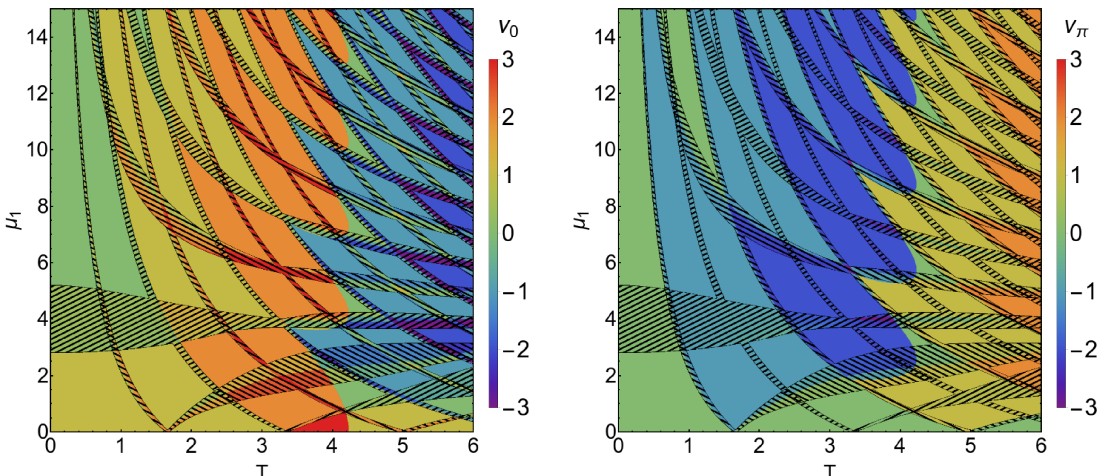

Figure 6: Winding numbers $\nu_0$ (left) and $\nu_\pi$ (right) in the $(T, \mu_1)$-plane. Hatched areas denote intermediate regions with exceptional points, either at $\beta = a_0$ (hatching: ///) or at $\beta = a_0 + i\pi/T$ (hatching: \\\). The parameters are: $J = 1$, $\gamma = 0.8$, $g = 1.0$, $\Delta = 0.3$, $\mu_2 = 0$, and $t_1 = t_2$.

of edge modes (blue lines) coming together and disappearing.

In order to better understand the behavior of the winding numbers, we show in Figure 6 a density plot of $\nu_0$ (left panel) and $\nu_\pi$ (right panel) in the $(T, \mu_1)$ plane. Regions with broken AUS are shown as hatched areas. We can see that, for $T$ slightly larger than 4, one of the winding numbers changes by 2 without an broken-AUS intermediate region. This supports the notion that there may be other mechanisms of AUS-breaking responsible for the noise around $\text{Re}(\beta) = a_0$ seen in Figure 5 for $T > 3$ which only appears under OBC. A better understanding of these phenomena will be left for future work.

Aside from the anomalous transitions described above, the phase diagram of the driven-dissipative Kitaev chain (as shown in Figure 6) has the same rib structure of Floquet resonances that appears in previous work [8,24,45]. The total number of edge modes, given by $|\nu_0| + |\nu_\pi|$, corresponds to that found by [45]. Although the splitting into $\nu_0$ and $\nu_\pi$ is clear from the viewpoint of topological classification, it is less obvious what the physical difference is between the two types of edge modes. It is not clear if these modes can be qualitatively distinguished by an appropriately chosen physical observable.

So far, for the driven-dissipative system, we have analyzed the quasi-rapidity spectrum and its link to topological phases. Presently, we will consider the impact of the transitions between these phases on the expectation values of observables in the Floquet steady state. These values can be computed by solving the following discrete-time Lyapunov equation (see Appendix F for further details):

$$\mathbf{C}_{F'} e^{\mathbf{X}_{F'} T} - e^{-\mathbf{X}_{F'}^\dagger T} \mathbf{C}_{F'} = i\mathbf{Q}. \tag{42}$$

Here $\mathbf{C}_{F'}$ is the stroboscopic covariance matrix in the infinite time limit at the first time-reversal invariant point, i.e. $t_0 = t'$. From this covariance matrix, we can derive stroboscopic expectation values of all other observables. Once again we consider the steady state current $\mathcal{J}$ under PBC, studied in section 3.4 for the undriven system. The results are shown in Figure 7 for two values of the system-bath coupling $g$. As we can appreciate, with weak dissipation (left panel), the rib-like resonance structure is clearly visible and there appears to be a direct cor-

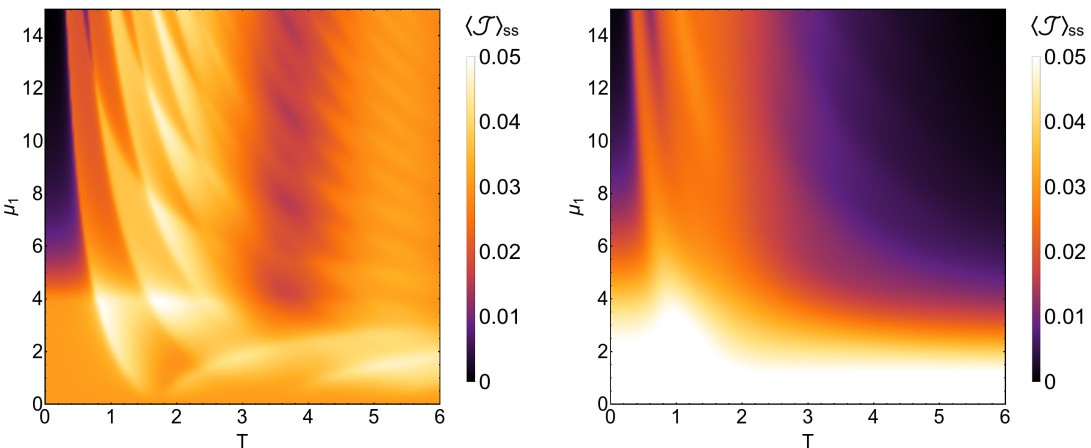

Figure 7: Floquet steady state expectation values of the current $\mathcal{J}$ under PBC. On the left panel (weak system-bath coupling $g = 0.1$), the current appears to be sensitive to the total winding number $|\nu_0| + |\nu_\pi|$. On the right panel, stronger dissipation ($g = 1.0$) causes the resonance structure to largely disappear. The remaining parameters are: $J = 1$, $\gamma = 0.8$, $\Delta = 0.3$, $\mu_2 = 0$, and $t_1 = t_2$.

respondence between the steady state current and the total winding number $|\nu_0| + |\nu_\pi|$. On the other hand, the stronger the dissipation the smoother the transitions become, so that for a strong enough dissipation this rib-like structure is barely visible (right panel).

## 5 Conclusions and outlook

By using tools from the study of non-Hermitian Hamiltonians, we have analyzed the topological properties of a driven-dissipative extension to the Kitaev chain. Dissipation is generated by an identical Markovian bath coupled locally to each site of the system, as described by a Lindblad master equation. By introducing a parameter $\Delta$ to the bath that interpolates between gain and loss, we can study a wide range of effects. For example, intermediate values of $\Delta$ break the chiral symmetry that protects topological order in the Kitaev chain, but create a new antiunitary symmetry (AUS) reminiscent of $\mathcal{PT}$ symmetry in non-Hermitian Hamiltonians. While this AUS leaves intact the different topological phases of the Kitaev chain, it introduces intermediate regions between the phases, where the AUS is broken and the winding number is no longer quantized. In these regions, the bulk-boundary correspondence breaks down.

The addition of periodic driving makes for some incredibly rich physics and we have only explored the tip of the iceberg. The phase diagram of the driven-dissipative system mostly follows the regions defined by Floquet resonances, which can overlap to produce higher winding numbers and multiple pairs of edge modes. Having multiple modes localized on each edge can lead to interesting phenomena such as hybridization. Precisely under which conditions these hybridized edge modes exist and what their relation is to the bulk topological order are still open problems. Like in the undriven open system, topological phases are separated by intermediate AUS-broken regions. However, there also seem to be other mechanisms of AUS-breaking, connected to the driving and the OBC, that are not yet understood.

The topological edge modes that we have studied, both in the driven and undriven case, are transient properties of the dissipative system. The presence of the bath leads the system to

a unique state at long times and bestows decay rates onto the edge modes. Interestingly, these decay rates can be different for the two edges of the chain, resulting in a left-right asymmetry that is reflected in the dynamics and steady state expectation values of various observables. Numerical results show some signatures of the topological order in the steady state, but an analytical connection has eluded us so far. Various ways to define winding numbers for density matrices have been proposed, for example using the Uhlmann phase [49], the interferometric phase suggested in [50], and more recently the Ensemble Geometric Phase [12]. While we stress that these mixed-state winding numbers are very different from the ones we have calculated for the band Liouvillian, it would be intriguing to apply them to the steady state of the dissipative Kitaev chain and compare this to our results.

Another promising angle for linking Floquet topological order to observables is borrowed from the study of spin chains and known as stroboscopic spin textures [26,45,51] or nonlocal string order [52,53], although it might not work with broken chiral symmetry. If applicable to our system, these techniques could potentially distinguish the two different winding number $\nu_0$ and $\nu_\pi$ in the driven system.

Finally, as potential experimental realizations of Majorana zero modes have been proposed as information processing devices to implement unitary gate operations in topological quantum computation [54], the generation of higher number Majorana modes using periodic systems opens the door to further explore in which way these new localized modes could optimally be used in quantum computation together to its robustness against the presence of external noise. Moreover, a thorough understanding of the decay times of Majorana modes under influence of the environment may prove important for future developments in this field.

# Acknowledgments

This work is part of the Delta-ITP consortium, a program of the Netherlands Organization for Scientific Research (NWO) that is funded by the Dutch Ministry of Education, Culture and Science (OCW) and of the research program of the Foundation for Fundamental Research on Matter (FOM), which is part of the Netherlands Organization for Scientific Research (NWO). The authors want to thank V. Gritsev, E. Ilievski, N. Robinson and J. van Wezel for guidance and interesting discussions.

# A   Geometric interpretation of the winding number

A $2 \times 2$ matrix with antiunitary symmetry given by (21) can only have the following form:

$$\mathbf{x}(k) = a_0 \mathbb{1} + i b(k) \sigma_x + i c(k) \sigma_y + d(k) \sigma_z , \tag{43}$$

where $a_0$, $b(k)$, $c(k)$ and $d(k)$ are real for all $k$. The matrix becomes defective when $b^2 + c^2 = d^2$ and this equation forms the surface of a double cone in the space spanned by $b$, $c$ and $d$. Inside of the cone, the antiunitary symmetry is spontaneously broken.

As $k$ traverses the Brillouin zone, $\mathbf{x}(k)$ defines a loop in this three-dimensional space, characterized by the system parameters such as $J$ and $\mu$. If the loop intersects the cone, then it does so at a pair of so-called exceptional points. Those are the points where the band gap closes. We can now define a topological invariant as the winding number of $\mathbf{x}(k)$ around this defective cone, interpreting it as an obstruction in $\mathbb{R}^3$. If the loop winds around the cone, it cannot be contracted without closing the gap (see Figure 8).

It is also clear from this geometric picture how $\gamma$ affects the transitions between the different phases. The loop defined by $\mathbf{x}(k)$ is an ellipse with semi-axes of length $2J$ and $2\gamma$. The

radius of curvature at its vertex ($k = 0$ or $\pi$) is $2\gamma^2/J$. If this is larger than $2g\Delta$, i.e. the radius of the cone, then the cone will touch the ellipse at its vertex when $|\mu| = 2J \pm 2g\Delta$. In the main text, we therefore require $|\gamma| > \sqrt{Jg\Delta}$. If $|\gamma|$ is smaller, the ellipse becomes too eccentric and will touch the cone for lower values of $|\mu|$. In that case one can have four exceptional points, as the ellipse intersects the cone in four different places. Furthermore, it is obvious that the ellipse cannot wind around the cone at all when $|\gamma| < g\Delta$, in which case the topologically nontrivial phase disappears entirely.

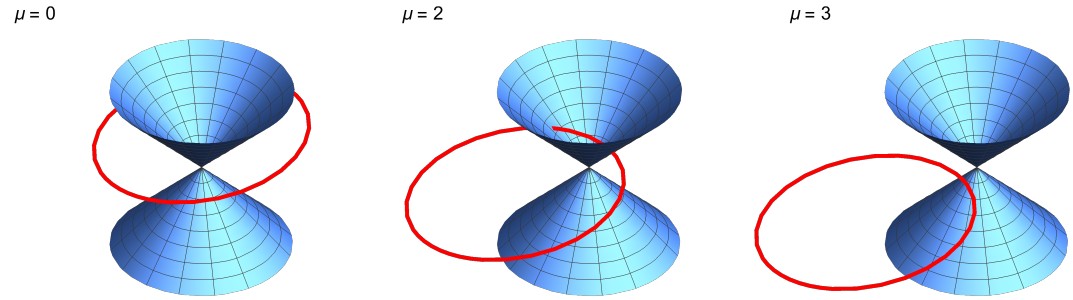

Figure 8: The space spanned by three of the free parameters of a two band matrix with AUS. The red loop represents $\mathbf{x}(k)$ as defined in eq. 19 for $k \in (0, \pi]$ and is drawn for three different values of $\mu$. The cone is the set of points in parameter space where the band gap closes. At $\mu = 2J$ in the intermediate regime (middle panel), $\mathbf{x}(k)$ intersects the cone at two exceptional points, at which it becomes defective.

## B  Computing the winding number

In order to compute the winding number of $\mathbf{x}(k)$, we define:

$$q(k) \equiv (2J\cos k + \mu) + 2i\gamma\sin k \equiv |q(k)|e^{-i\theta(k)}, \quad \tan\xi(k) = \frac{2ig\Delta}{|q(k)|}. \qquad (44)$$

The left and right eigenvectors of $\mathbf{x}(k)$ can then be written as:

$$\underline{u}_1(k) = \begin{pmatrix} e^{-i\theta(k)}\cos\frac{\xi(k)}{2} \\ \sin\frac{\xi(k)}{2} \end{pmatrix}, \qquad \underline{u}_2(k) = \begin{pmatrix} -e^{-i\theta(k)}\sin\frac{\xi(k)}{2} \\ \cos\frac{\xi(k)}{2} \end{pmatrix}, \qquad (45)$$

$$\underline{v}_1(k) = \begin{pmatrix} e^{-i\theta(k)}\cos^*\frac{\xi(k)}{2} \\ \sin^*\frac{\xi(k)}{2} \end{pmatrix}, \qquad \underline{v}_2(k) = \begin{pmatrix} -e^{-i\theta(k)}\sin^*\frac{\xi(k)}{2} \\ \cos^*\frac{\xi(k)}{2} \end{pmatrix}, \qquad (46)$$

such that $\underline{v}_i^* \cdot \underline{u}_j = \delta_{ij}$. The two winding numbers (24) associated with these bands are:

$$\nu = \frac{1}{2\pi}\int_{-\pi}^{\pi}(1 \pm \cos\xi)\,\mathrm{d}\theta = \frac{1}{2\pi}\int_{-\pi}^{\pi}\left(1 \pm \frac{2ig\Delta}{\sqrt{|q(k)|^2 - 4g^2\Delta^2}}\right)\frac{\partial\theta}{\partial k}\,\mathrm{d}k. \qquad (47)$$

Note that the first term $\oint \mathrm{d}\theta$ is simply the winding number of $q(k)$ around the origin in the complex plane, which corresponds to the winding number of the closed Kitaev chain. The second term only contributes to the real part of $\nu$ in the regime where $\mathcal{PT}$ symmetry is broken. We can further simplify:

$$\frac{\partial\theta}{\partial k} = \frac{4J\gamma\sin^2 k}{|q(k)|^2} + \frac{2\gamma\cos k(2J\cos k + \mu)}{|q(k)|^2} = \frac{2\gamma(2J + \mu\cos k)}{|q(k)|^2}. \qquad (48)$$

The numerical results of the integral are shown in Figure 1(d).

## C  Topology of X vs. A

The matrix $\mathbf{X}$ does not give all the necessary information about the full Liouvillian time evolution, despite determining the entire spectrum. The decay modes and the steady state also depend on the off-diagonal block $\mathbf{Y}$. To build a full description of the time evolution, we also need to know the eigenvectors of $\mathbf{A}$, which form the so-called *normal master modes*, superoperators that map one decay mode into another. These eigenvectors can be build up from the eigenvectors of $\mathbf{X}$ and the steady state covariance matrix $\mathbf{C}$, which in turn depends on $\mathbf{Y}$. To see this, we can write:

$$\mathbf{A} = \begin{pmatrix} -\mathbf{X}^\dagger & -i\mathbf{Y} \\ 0 & \mathbf{X} \end{pmatrix} = \mathbf{W}^{-1} \begin{pmatrix} -\mathbf{X}^\dagger & 0 \\ 0 & \mathbf{X} \end{pmatrix} \mathbf{W}, \tag{49}$$

$$\mathbf{W} = \begin{pmatrix} \mathbb{1} & \mathbf{C} \\ 0 & \mathbb{1} \end{pmatrix}, \qquad \mathbf{X}^\dagger \mathbf{C} + \mathbf{C}\mathbf{X} = i\mathbf{Y}. \tag{50}$$

Therefore the left and right eigenvectors of $\mathbf{A}$, denoted by $\underline{\phi}_i^\pm$ and $\underline{\psi}_i^\pm$ respectively, can be written as:

$$\mathbf{A}\underline{\psi}_i^\pm = \pm\beta_i\underline{\psi}_i^\pm, \quad \mathbf{A}^\dagger\underline{\phi}_i^\pm = \pm\beta_i^*\underline{\phi}_i^\pm, \qquad \underline{\phi}_i^{\mu*} \cdot \underline{\psi}_j^\nu = \delta_{ij}\delta_{\mu\nu},$$

$$\underline{\psi}_i^+ = \mathbf{W}^{-1}\begin{pmatrix} 0 \\ \underline{u}_i \end{pmatrix} = \begin{pmatrix} -\mathbf{C}\underline{u}_i \\ \underline{u}_i \end{pmatrix}, \qquad \underline{\psi}_i^- = \mathbf{W}^{-1}\begin{pmatrix} \underline{v}_i \\ 0 \end{pmatrix} = \begin{pmatrix} \underline{v}_i \\ 0 \end{pmatrix}, \tag{51}$$

$$\underline{\phi}_i^+ = \mathbf{W}^T\begin{pmatrix} 0 \\ \underline{v}_i \end{pmatrix} = \begin{pmatrix} 0 \\ \underline{v}_i \end{pmatrix}, \qquad \underline{\phi}_i^- = \mathbf{W}^T\begin{pmatrix} \underline{u}_i \\ 0 \end{pmatrix} = \begin{pmatrix} \underline{u}_i \\ \mathbf{C}^T\underline{u}_i \end{pmatrix},$$

where $\underline{v}_i$ and $\underline{u}_i$ are the left and right eigenvectors of $\mathbf{X}$, as given in (22). The right eigenvectors $\underline{\psi}_i^\pm$ provide the normal master modes of eq. (27). All of the above can be written in Fourier space, as functions of the quasimomentum $k$, and this allows us to compute the winding numbers for the four bands of the matrix $\mathbf{A}(k)$:

$$v_i^+ = \frac{i}{\pi}\int_{-\pi}^{\pi} \mathrm{d}k\ \underline{\phi}_i^{+*}(k) \cdot \frac{\partial}{\partial k}\underline{\psi}_i^+(k) = \frac{i}{\pi}\int_{-\pi}^{\pi} \mathrm{d}k\ \underline{v}_i^*(k) \cdot \frac{\partial}{\partial k}\underline{u}_i(k) = v_i, \tag{52}$$

$$v_i^- = \frac{i}{\pi}\int_{-\pi}^{\pi} \mathrm{d}k\ \underline{\phi}_i^{-*}(k) \cdot \frac{\partial}{\partial k}\underline{\psi}_i^-(k) = \frac{i}{\pi}\int_{-\pi}^{\pi} \mathrm{d}k\ \underline{u}_i^*(k) \cdot \frac{\partial}{\partial k}\underline{v}_i(k) = v_i^*, \tag{53}$$

where $v_i$ are the winding numbers associated with the bands of $\mathbf{x}(k)$, as given by eq. (24). Integration by parts shows that the two results are complex conjugates. This provides a thorough justification to only study the topological band structure of $\mathbf{x}(k)$.

## D  Edge modes of X

One way to prove the presence of localized edge modes is to consider the linearized Dirac form of $\mathbf{x}(k)$ around $k = 0$:

$$\mathbf{x}(k) \approx g(1+\Delta^2) + 2i\gamma k\sigma_x - i(2J+\mu)\sigma_y + 2g\Delta\sigma_z. \tag{54}$$

A Fourier transform to real space yields

$$\mathbf{x}(r) \approx g(1+\Delta^2) - 2\gamma\sigma_x\frac{\partial}{\partial r} + im\sigma_y + 2g\Delta\sigma_z, \tag{55}$$

where we have defined the Dirac mass $m \equiv -2J - \mu$. Now we consider $m$ to vary in space, such that $m(r) = m_1 \Theta(-r) + m_2 \Theta(r)$, with $\Theta(r)$ the Heaviside step function. An eigenvector $\underline{u}(r)$ with eigenvalue $\beta$ must satisfy:

$$\frac{\partial}{\partial r}\underline{u}(r) = \frac{1}{2\gamma}\left(\delta\beta\,\sigma_x - 2ig\Delta\sigma_y - m(r)\sigma_z\right)\underline{u}(r) \equiv \mathbf{B}(r)\underline{u}(r), \tag{56}$$

with $\delta\beta \equiv g(1 + \Delta^2) - \beta$. The solution of this differential equation is of the form $\underline{u}(r) = \exp\left[\int_0^r \mathrm{d}r'\mathbf{B}(r')\right]\underline{u}(0)$. A localized solution is only possible when the eigenvalues of $\mathbf{B}(r)$ have a real part that changes sign at $r = 0$, such that $\underline{u}(r)$ falls off exponentially on either side of the domain wall. This is the case when $\delta\beta = \pm 2g\Delta$ and $m_1$ and $m_2$ have different signs. Since changing the sign of $m$ puts us in a different topological phase, this proves that an edge mode appears on the boundary between two phases[2]. The eigenvalues corresponding to these localized solutions are:

$$\beta = g(1 + \Delta^2) \pm 2g\Delta = g(1 \pm \Delta)^2, \tag{57}$$

where the sign depends on the orientation of the domain wall.

Finally, note that we can model the vacuum as a fermionic chain with $\mu \to -\infty$, placing it firmly in the topologically trivial regime (as expected). We can therefore use this same analysis to describe edge modes of the Kitaev chain with open boundary conditions. As long as $|\mu| < 2J$, each edge forms a domain wall and allows for a localized edge mode.

## E   Time evolution of observables

The time evolution of the covariance matrix $\mathbf{C}(t)$ can be expressed in terms of the eigenvalues and eigenvectors of $\mathbf{X}$. The covariance matrix satisfies the following differential matrix equation [8]:

$$\frac{\mathrm{d}}{\mathrm{d}t}\mathbf{C}(t) = -\mathbf{X}^T\mathbf{C}(t) - \mathbf{C}(t)\mathbf{X} + i\mathbf{Y}. \tag{58}$$

Assuming $\mathbf{X}$ is diagonalizable and that the steady state is unique, we find as a solution:

$$\mathbf{C}(t) = \mathbf{C}_{ss} + \frac{1}{2}\sum_{i,j}e^{-t(\beta_i + \beta_j)}\left(\underline{v}_i \otimes \underline{v}_j - \underline{v}_j \otimes \underline{v}_i\right)^*\left(\underline{u}_i \cdot (\mathbf{C}_0 - \mathbf{C}_{ss})\underline{u}_j\right), \tag{59}$$

where $\mathbf{C}_{ss}$ is the covariance matrix in the steady state, found by solving eq. (13) (with $\mathbf{C}_{ss} = \mathbf{C}$), and $\mathbf{C}_0$ is the covariance matrix for the initial state. In the limit $t \to 0$, we recover $\mathbf{C}(0) = \mathbf{C}_0$ by identifying two resolutions of the identity in our biorthogonal basis. If the initial state is a pure energy eigenstate (e.g. the ground state), we can compute $\mathbf{C}_0$ from the eigenvectors of the $2N \times 2N$ matrix $\mathbf{H}$:

$$\mathbf{C}_0 = 2i\sum_{n=1}^{n_f}\mathrm{Im}\left(\underline{\psi}_n^* \otimes \underline{\psi}_n\right), \tag{60}$$

with $\mathbf{H}\underline{\psi}_n = \epsilon_n\underline{\psi}_n$ and $\epsilon_n \leq \epsilon_{n+1}$. For the ground state, we set $n_f = N$ as the Fermi level. In the topologically nontrivial phase, the Kitaev chain has a degenerate ground state with $\epsilon_N = \epsilon_{N+1} = 0$. We can then choose $n_f = N - 1$ or $n_f = N + 1$, depending on whether the edge zero-mode is filled or not.

---

[2]The same analysis can be done around $k = \pi$, in order to find the phase transition at $\mu = +2J$.

# F  Off-diagonal Floquet blocks and their importance

In order to compute the off-diagonal block $\mathbf{Q}$ in eq. (36), we first write:

$$e^{\mathbf{A}_i t_i} = \begin{pmatrix} e^{-\mathbf{X}_i^\dagger t_i} & i\mathbf{Z}_i \\ 0 & e^{\mathbf{X}_i t} \end{pmatrix}, \tag{61}$$

where $i = 1, 2$ and $\mathbf{Z}_i$ must satisfy the Lyapunov equation

$$\mathbf{X}_i^\dagger \mathbf{Z}_i + \mathbf{Z}_i \mathbf{X}_i = \mathbf{Y} \exp(t_i \mathbf{X}_i) - \exp(-t_i \mathbf{X}_i^\dagger) \mathbf{Y}, \tag{62}$$

as is easily shown by requiring that $[\mathbf{A}_i, \exp(\mathbf{A}_i t_i)] = 0$. A formal solution can be obtained iteratively, but this is not very illuminating.

Now, by simple matrix multiplication, $\mathbf{Q}$ takes the form:

$$\mathbf{Q} = e^{-\mathbf{X}_1^\dagger t_1/2} e^{-\mathbf{X}_2^\dagger t_2} \mathbf{Z}_1 + \mathbf{Z}_1 e^{\mathbf{X}_2 t_2} e^{\mathbf{X}_1 t_1/2} + e^{-\mathbf{X}_1^\dagger t_1/2} \mathbf{Z}_2 e^{\mathbf{X}_1 t_1/2}. \tag{63}$$

While this expression seems intractable, it can be evaluated numerically. The importance of this matrix becomes clear when we compute the stroboscopic time evolution of the covariance matrix. We start with eq. (58), which also holds when $\mathbf{X}$ is time-dependent. This equation can be rewritten using the structure matrix $\mathbf{A}(t)$ as

$$\frac{\mathrm{d}}{\mathrm{d}t} \mathbf{D}(t) = [\mathbf{A}(t), \mathbf{D}(t)], \quad \mathbf{D}(t) \equiv \begin{pmatrix} \mathbb{1} & \mathbf{C}(t) \\ 0 & 0 \end{pmatrix}, \tag{64}$$

which in turn is satisfied by the following form:

$$\mathbf{D}(t) = \mathcal{T} \exp\left( \int_0^t \mathrm{d}\tau \mathbf{A}(\tau) \right) \mathbf{D}(0) \, \mathcal{T} \exp\left( -\int_0^t \mathrm{d}\tau \mathbf{A}(\tau) \right). \tag{65}$$

We can now find a solution for the Floquet steady state by requiring that $\mathbf{D}$ is unchanged after one period:

$$\mathbf{D}_F = e^{\mathbf{A}_F T} \mathbf{D}_F \, e^{-\mathbf{A}_F T} \quad \Rightarrow \quad \mathbf{C}_F e^{\mathbf{X}_F T} - e^{-\mathbf{X}_F^\dagger T} \mathbf{C}_F = i\mathbf{Q}, \tag{66}$$

where eq. (36) was used to obtain a *discrete-time Lyapunov equation* for the Floquet steady state covariance matrix $\mathbf{C}_F$ [8].

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
