# Peer review of "Dynamical signatures of topological order in the driven-dissipative Kitaev chain"

_SciPost Physics, doi:SciPost Phys. 6, 026 (2019)_

## Round 1 · Referee Report · Anonymous · 2018-12-19

Strengths

1) Originality

Weaknesses

1) Not so general result

Report

The paper is interesting. The authors take the Kitaev chain, they add dissipation so that the system is still quadratic in fermionic operators (and then integrable) and construct a generalization of the winding number to study the topological properties. They still find a topological phase with edge modes which decay in time, and a normal phase. There is also an intermediate phase with no topology and gapless imaginary part of the rapidities. The topological phase can be experimentally seen in the properties of the steady-state current. The authirs also extend this picture to driven-dissipative systems finding evidence of non-trivial winding numbers in the Floquet eigenmode bands (imaginary parts of the Floquet quasi-rapidities). The paper is worth of publication in SciPost. I have just some comments
1) Is there some dependence of the winding number Eq.26 on the choice of the rapidity band where it is evaluated?
2) Just a curiosity. How does the width of the intermediate region with gapless imaginary part of the rapidity depend on the dissipation strength? Is there a value of the dissipation strength where the gapped windows in Figs. 2-right and 5-bottom close up? At that point, I imagine that the systems shows no topological properties at all. How does this reflects in the behaviour of the current?
3) There are small typos: p.8 ``pointss'', p.15 ``more that two'', p.16 ``Brillioun''

Requested changes

1) In the abstract the authors write about ``powerful and widely applicable approach to topology in driven-dissipative systems''. If I understand correctly the approach they describe is strictly restricted to quadratic fermionic Hamiltonians. The authors should clarify why their approach is ``widely applicable'' or withdraw this sentence.

  • validity: good
  • significance: ok
  • originality: good
  • clarity: good
  • formatting: good
  • grammar: good

Author:  Moos van Caspel  on 2019-01-28  [id 417]

(in reply to Report 1 on 2018-12-19)
Category:
answer to question

Thank you for your kind comments. Below, we try to answer the questions your raised.

1) There is indeed a dependence on the choice of rapidity band. However, we note that $\sum_i \nu_i \mod 2 = 0$. For the purpose of defining topological phases and the bulk-boundary correspondence, it is therefore sufficient to focus on one band. We have added a line below Eq.26 to clarify this.

2) Yes, the width of the intermediate region scales linearly with the dissipation strength $g$, as described in section 3.1 for the undriven system. With driving, this is not simple to see analytically. The referee correctly remarks that the topologically nontrivial phase vanishes for sufficiently strong dissipation. The paper assumes a relatively weak dissipation (see footnote on page 7), which is consistent with the Born-Markov approximation that underlies the Lindblad master equation. The steady state current's dependence on $\mu$ (figure 4 bottom panels, and figure 7) will be further smoothened out if the dissipation increases, but shows no novel behavior when the gapped windows close.

3) The typos have been corrected. The last one appeared in more than one place but we have made sure that it has been amended in all places.

---

## Round 1 · Referee Report · Anonymous · 2019-1-4

Strengths

Completeness and clarity

Weaknesses

Free Lindbladian models

Report

The manuscript studies a new concept in condensed matter and quantum physics: open topological quantum matter. This is a timely article due to the fact that in the past it has been studied the concept of topological order or the dynamics of driven and dissipative quantum systems. Nonetheless, in recent years, and motivated by new experimental results, new concepts, fields or setups have appeared. (see for instance: Nature Physics volume 4, pages 878–883 (2008) and Nature Physics volume 7, pages 971–977 (2011)).

The article is clearly written and complete in the discussion. It discusses the different dynamical elements that appear in a free Lindbladian evolution. Nonetheless, this part has already appeared in the literature (see for instance: Phys. Rev. Lett. 109, 130402 (2012) and New J. Phys. 15, 085001 (2013)).

One of the new results is to define the winding number in a "biorthogonal basis". Here, a question appears, how different or how similar is this measure of topological order compare with the one given in Phys. Rev. Lett. 112, 130401 (2014)?

Requested changes

Could the authors clarify their definition of the winding number comparing with previous ones?

  • validity: good
  • significance: good
  • originality: good
  • clarity: good
  • formatting: good
  • grammar: good

Author:  Moos van Caspel  on 2019-01-28  [id 416]

(in reply to Report 2 on 2019-01-04)
Category:
answer to question

  • We thank the referee for pointing out these two references. We have cited the very relevant second paper at the start of section 3 in our resubmission. While we understand that the first paper suggested is a pioneering work in the field, we believe it does not relate directly to our submission.

  • The two references given (Phys. Rev. Lett. 109, 130402 (2012) and New J. Phys. 15, 085001 (2013)) take an approach that is very different from ours. To the best of our understanding, they aim to construct fully dissipative systems (with no intrinsic Hamiltonian) in which the steady state(s) exhibits some form of topological order. Our paper, on the hand, seeks to understand the dynamical behavior of a known Hamiltonian system with topological order once dissipation is added. See also our reply to the next question. It is conceivable that the decaying edge modes we find are somehow related to the zero-purity modes of New J. Phys. 15, 085001 (2013), which would be an interesting direction for future work. We have added these references at the start of section 3.

  • This is an interesting point. The Uhlmann phase from the cited paper is used to characterize topological order for mixed states which result when the system is coupled to a thermal bath in the "traditional" way: by the thermal ensemble. Firstly, this approach has been shown in \url{https://doi.org/10.1098/rsta.2015.0231} to lead to inconsistencies in the Kitaev chain at finite temperature. Secondly, while techniques to study the topology of mixed states can be applied to the steady state of an open quantum system, and can possibly lead to interesting new types of topological order, there is no relation to the topological order of the original, closed system (tied to the ground state) if the steady state is unique. Because we are interested in the effect of dissipation on topological order of the Kitaev chain, we use an entirely different approach in which a winding number is associated with the generator of the nonunitary time evolution, rather than with any one mixed quantum state. This leads to the dynamical signatures we described, including edge modes with exponential decay.

In our case, the steady state is unique and does not contain (zero-damping) Majorana modes, as we argued in the paper. Thus, we expect that the application of the Uhlmann phase would always render a trivial topological phase. It would be indeed interesting to confirm our intuition, but this falls beyond the scope of the present paper. We have expanded upon this in the conclusions and outlook section, also including the corresponding reference.

---

## Round 2 · Referee Report · Anonymous · 2019-1-29

Strengths

Completeness and clarity

Weaknesses

Free Lindbladian models

Report

After reading the second version of the manuscript, we recommend the publication of the article

---

## Round 2 · Author Response

We would like to thank both referees for their comments. In the present revised version of the manuscript we have tried to accommodate their comments as best as we could.

---

## Round 2 · List of Changes

- We have retracted the final sentence of the abstract.
- We have added a sentence just before the last paragraph of section 3.1, explaining why we focus on only one of the rapidity bands.
- We have added several lines to the conclusion of our paper, in order to more clearly relate our definitions and our results to the existing literature.
- We have added multiple references at the start of section 3, as kindly suggested by the referees.
- At the start of section 4.3, we have changed the chemical potential of the time-averaged Hamiltonian, relaxing the assumption that $t_1 = t_2$.
- We have amended various typographical errors.

---

## Editorial Decision

published